# Targeting Triple Negative Breast Cancer Stem Cells by Heat Shock Protein 70 Inhibitors

**DOI:** 10.3390/cancers14194898

**Published:** 2022-10-07

**Authors:** Chia-Hung Tsai, Jing-Ru Weng, Hsiang-Wen Lin, Meng-Tien Lu, Yu-Chi Liu, Po-Chen Chu

**Affiliations:** 1Department of Surgery, Taichung Tzu Chi Hospital, Buddhist Tzu Chi Medical Foundation, Taichung 427213, Taiwan; 2Department of Marine Biotechnology and Resources, National Sun Yat-sen University, Kaohsiung 804201, Taiwan; 3Institute of BioPharmaceutical Sciences, National Sun Yat-sen University, Kaohsiung 804201, Taiwan; 4School of Pharmacy, College of Pharmacy, China Medical University, Taichung 406040, Taiwan; 5Department of Pharmacy, China Medical University Hospital, Taichung 406040, Taiwan; 6Department of Cosmeceutics and Graduate Institute of Cosmeceutics, China Medical University, Taichung 406040, Taiwan

**Keywords:** triple negative breast cancer, Hsp70 inhibitor, breast cancer stem cells, mammosphere, β-catenin

## Abstract

**Simple Summary:**

Triple negative breast cancer (TNBC) has been a leading cause of cancer deaths among women and limited treatment options are currently available. Consequently, developing the promising therapy for TNBC remains an unmet medical need. The aim of this study was to assess the effect of Hsp70 inhibitors on breast cancer stem cells (BCSCs) in TNBC cells. The present study demonstrated the potent in vitro and in vivo anti-BCSCs activity of two Hsp70 inhibitors, compound 1 and 6, in TNBC cells. Our findings suggest that targeting Hsp70 could be an attractive strategy for eliminating BCSCs populations in TNBC.

**Abstract:**

Triple negative breast cancer (TNBC) is considered the most aggressive breast cancer with high relapse rates and poor prognosis. Although great advances in the development of cancer therapy have been witnessed over the past decade, the treatment options for TNBC remain limited. In this study, we investigated the effect and potential underlying mechanism of the Hsp70 inhibitors, compound 1 and compound 6, on breast cancer stem cells (BCSCs) in TNBC cells. Our results showed that compound 1 and 6 exhibited potent tumor suppressive effects on cell viability and proliferation, and effectively inhibited BCSC expansion in TNBC cells. Reminiscent with the effect of Hsp70 inhibitors, Hsp70 knockdown effectively suppressed mammosphere formation and the expressions of BCSCs surface markers. Mechanistically, evidence showed that the Hsp70 inhibitors inhibited BCSCs by down-regulating β-catenin in TNBC cells. Moreover, we used the Hsp70 inhibitors treated TNBC cells and a stable Hsp70 knockdown clone of MDA-MB-231 cells to demonstrate the in vivo efficacy of Hsp70 inhibition in suppressing tumorigenesis and xenograft tumor growth. Together, these findings suggest the potential role of Hsp70 as a target for TNBC therapy and foster new therapeutic strategies to eliminate BCSCs by targeting Hsp70.

## 1. Introduction

Triple negative breast cancer (TNBC) is a subset of breast cancer, which is deficient in the expression of estrogen receptor (ER), progesterone receptor (PR), and human epidermal growth factor receptor 2 (HER2); therefore, TNBC does not respond to hormonal and anti-HER2 antibody therapies [1]. TNBC accounts for 15–20% of all breast cancer cases. TNBC is usually more aggressive, with frequent metastasis, and has a relatively poor clinical outcome and higher grade than other types of breast cancer [2]. The massive heterogeneity of TNBC and the lack of defined molecular targets make it challenging to treat.

Cancer stem cells (CSCs), alternatively tumor-initiating cells, are the small subpopulation within cancer cells possessing the capacity to self-renew and to recapitulate the heterogeneity of the original tumor, which is one of the reasons contributing to cancer treatment failure and disease progression [3]. Moreover, accumulating evidence indicates that CSCs are responsible for tumor initiation, recurrence, metastasis, prognosis, and resistance to conventional therapy [4,5]. Breast cancer stem cells (BCSCs) are defined as a subset of breast cancer cells exhibiting the properties of self-renewal and differentiation capacity, which are thought to contribute to tumor relapse and metastasis [6]. The cell surface markers CD44^+^/CD24^−^ and high aldehyde dehydrogenase (ALDH) activity have been established as biomarkers for BCSCs isolation and characterization in vitro and in vivo [7,8]. BCSCs are the leading cause of tumor progression and therapy resistance that results in poor patient outcomes [9]. Therefore, selective targeting of BCSCs is a promising therapeutic strategy to abolish the progression of breast cancer, reduce the risk of recurrence, and improve patient outcome [10]. Additionally, aberrant activation of the Wnt/β-catenin, Hedgehog, and Notch signaling pathways has been implicated in regulating BCSCs [11,12,13]. Accordingly, targeting these pathways to eliminate BCSCs might have therapeutic potential in breast cancer treatment.

Heat shock proteins (HSPs) are a large family of molecular chaperones that are crucial in diverse cellular processes by participating in DNA repair processes and acting as regulators of protein homeostasis, including folding, assembly/maturation, secretion, transportation, translocation, and stability [14,15]. The stress-inducible heat shock protein 70 (Hsp70) participates in the protection of proteins from aggregation and promotes refolding and degradation of damaged polypeptides [16,17]. Overexpression of Hsp70 has been observed in multiple forms of cancers, including breast, colon, liver, prostate, esophagus, and cervix [18,19,20,21,22,23], and correlates with tumor grade, metastasis, chemotherapy resistance and poor prognosis [19]. Moreover, elevated expression of Hsp70 has been shown to increase the tumorigenicity of cancer [24] and protect cancer cells from apoptosis in response to a variety conditions of survival stresses, for example, hypoxia, nutrient deprivation or anticancer treatment [25,26,27]. Hence, Hsp70 has emerged as a promising target for anticancer drug development. Indeed, various Hsp70 inhibitors have shown potent effectiveness and are currently in preclinical evaluation [28,29]. Since inactivation of the *Hsp70* gene has been found to inhibit the tumorigenesis, invasion and metastasis of mammary tumors in vivo [30], targeting Hsp70 could be an attractive strategy for eliminating BCSCs populations in breast cancer.

Previously, based on the screening of a novel series of rhodacyanine-based Hsp70 inhibitors, we identified that compound 1 and compound 6 exhibited high potency to inhibit Hsp70’s chaperone activity and antiproliferative activities against breast cancer cells [31]. Equally important, in contrast to the reported effect of Hsp90 inhibitors on Hsp70 upregulation, compounds 1 and 6 did not cause a compensatory elevation in Hsp90 expression [31]. Although the pre-clinical developments of Hsp70 inhibitors as a single agent or in combination with other drugs in cancer therapy are ongoing [28,32], the direct effect of Hsp70 inhibitors in targeting BCSCs remains unknown. The present study aimed to elucidate the effect of the Hsp70 inhibitors, compound 1 and compound 6, on BCSCs and the underlying mechanism of Hsp70 inhibition for maintaining breast cancer stemness in TNBC cells. Our results demonstrated that compounds 1 and 6 exhibited not only potent cytotoxicity and anti-proliferative activity against TNBC cells, but also high efficacy in inhibiting BCSCs characteristics, such as mammosphere-forming capacity and CSC-associated markers. Reminiscent with the effect of compounds 1 and 6 on BCSCs, genetic knockdown of Hsp70 also resulted in inhibition of BCSCs in TNBC MDA-MB-231 cells. Mechanistically, β-catenin is identified to be the potential target of Hsp70 inhibitors for maintaining the stemness properties of TNBC cells. Moreover, Hsp70 inhibition in suppressing tumorigenesis and xenograft tumor growth were further confirmed in vivo using Hsp70 inhibitors treated TNBC cells and a stable Hsp70 knockdown clone of MDA-MB-231 cells, respectively. Together, our findings provide and foster a novel anti-TNBC strategy to eliminate BCSCs by targeting Hsp70.

## 2. Materials and Methods

### 2.1. Cell Culture and Antibodies

Human breast cancer cell lines, MCF-7, T47D, MDA-MB-231, MDA-MB-468, HCC-1937, HCC-1806 and MCF-10A human mammary epithelial cells were obtained from the American Type Culture Collection (Manassas, VA, USA). These cell lines were maintained in recommended growth medium with 10% fetal bovine serum (FBS; Invitrogen, Carlsbad, CA, USA) at 37 °C in a humidified incubator containing 5% CO_2_. The antibodies used in this study and their sources were as follows: Hsp70, Hsp90, β-Actin, β-catenin (Santa Cruz Biotechnology, Santa Cruz, CA, USA); KLF4, NF-κB, Cyclin D1, Notch1 (Cell Signaling, Beverly, MA, USA); CD133, CD44 (GeneTex, Irvine, CA, USA).

### 2.2. Cell Viability Assay

The effects of the compound 1 and compound 6 on cell viability were determined by 3-(4,5-dimethylthiazol-2-yl)-2,5-diphenyltetrazolium bromide (MTT) assays. Cells were seeded in 96-well plates at a density of 5000 cells per well in the presence of 10% FBS. After overnight incubation, cells were exposed to test agents vis-à-vis vehicle in the presence of 5% FBS for 72 h. After treatment, cells were incubated with MTT (Biomatik, Wilmington, DE, USA) for an additional 1 h. The medium was then removed from each well and replaced with DMSO for subsequent colorimetric measurement of absorbance at 570 nm. Cell viabilities are expressed as percentages of viable cells relative to the corresponding vehicle-treated control group.

### 2.3. Colony Formation Assay

Cells were seeded in six-well plates at a density of 2 × 10^3^ to 1 × 10^4^ cells per well, left to attach overnight, and then incubated with media containing compound 1 or compound 6 for 7–14 days until colonies were visible. The colonies were fixed with 4% formaldehyde (Sigma-Aldrich, St. Louis, Mo, USA) and stained with crystal violet (5 mg/mL in 2% ethanol; Sigma-Aldrich). Colonies containing more than 50 cells were counted. Cell proliferation was determined from the numbers of colonies and expressed as a percentage relative to the corresponding vehicle-treated control group.

### 2.4. Mammosphere Formation Assay

A total of 2 × 10^3^ MDA-MB-231 and 1 × 10^4^ MDA-MB-468 cells/well were seeded in ultra-low attachment 24-well plates (Corning Inc., Union City, CA, USA) and maintained in MammoCult™ Human Medium (STEMCELL Technologies; Vancouver, BC, Canada) with test compounds. After 8–10 days, the formation of mammosphere was determined and expressed as percentage of CSCs relative to the corresponding vehicle-treated control group.

### 2.5. Stable Hsp70 Knockdown with shRNA Lentivirus

All of the lentiviral vectors and shRNAs against Hsp70 were obtained from the National RNAi Core Facility (Academia Sinica, Taipei, Taiwan). The lentivirus particles carrying control shRNA and Hsp70 shRNAs were generated by co-transfecting shRNA plasmids with lentiviral packaging plasmids pSPAX2 and pMD2.G into 293T cells for 72 h. Stable knockdown cell clones were established by infecting MDA-MB-231 cells with shRNA lentivirus followed by puromycin selection for 7–10 days.

### 2.6. Cell Migration Assay

The transwell cell migration assay using transwell chambers (8 µm pore-size, Millipore, Bedford, MA, USA) was performed to examine the migration ability of cells. Plate 3 × 10^4^ cells in 300 μL of serum-free medium into the upper chambers of each transwell inserts in triplicate and then add 800 μL of medium with 10% FBS (Invitrogen) into the lower chambers. After 16–24 h incubation, the chambers were fixed in methanol and stained with Giemsa. The cells that migrated through the pores to the lower surface of the inserts were counted using a light microscope at 100× magnification in randomly selected fields of each independent experiment.

### 2.7. Plasmid and Transient Transfection

The human β-catenin plasmid was obtained from Addgene (Addgene plasmid #16518). Transient transfections were performed using Lipofectamine 2000 (Invitrogen) according to the manufacturer’s protocol.

### 2.8. Western Blot

Total protein lysates were collected and solubilized in lysis buffer (25 mM Tris-HCl pH 7.5, 150 mM NaCl, 1% NP-40, 1% sodium deoxycholate, 0.1% SDS) supplemented with protease inhibitors on ice for 30 min and then sonicated for 1 min. Protein concentration was determined by using BCA Protein Assay kit (Thermo Fisher Scientific, Waltham, MA, USA). Total proteins were separated by SDS-PAGE and then transferred onto nitrocellulose membranes. After blocking in 5% non-fat milk for 1 h, the membranes were incubated overnight at 4 °C with primary antibodies. Horseradish peroxidase (HRP)-conjugated goat anti-rabbit or anti-mouse antibody was applied as a secondary antibody for 1 h at room temperature. Protein detection was performed by Chemiluminescence Reagent Plus (Perkin-Elmer, Waltham, MA, USA). The relative intensity of the protein band was measured densitometrically using the Image J software. Original western blot images are found in Appendix A. 

### 2.9. In Vivo Study

All experimental procedures using mice were performed in accordance with the relevant guidelines and regulations and approved by the Institutional Animal Care and Use Committee (IACUC) of China Medical University. For the tumorigenicity model, DMSO, compound 1, or compound 6 treated MDA-MB-231 cells (1 × 10^3^ living cells in 0.1 mL; 50% Matrigel in PBS) were subcutaneously injected into both flanks of the 8-week-old athymic nude mice obtained from National Laboratory Animal Center (NLAC, Taipei, Taiwan) (*n* = 6). After 12 weeks post-injection, the tumor incidence in mice receiving compound 1 or compound 6 treated cells was compared to the control DMSO group. For the xenograft tumor growth model, stable clone of control (Scrb) or Hsp70 knockdown (shHsp70 #755) MDA-MB-231 cells (1 × 10^6^ cells in 0.1 mL; 50% Matrigel in PBS) was subcutaneously injected in 8-week-old athymic nude mice (*n* = 4). Tumor volumes were calculated weekly from caliper measurements (with^2^ × length × 0.52).

### 2.10. Statistical Analysis

The experiments in this study were performed using 3–6 replicates in each group. Most data are expressed as the mean ± standard deviation (S.D.) from at least two experiments. Differences in the effects of various treatments were compared using Student’s *t*-test. Differences were considered significant at *p* < 0.05.

## 3. Results

### 3.1. Hsp70 Expression Is Elevated in Breast Tumor Tissue and Negatively Correlated with Breast Cancer Prognosis

Hsp70 has been reported to be overexpressed and associated with enhanced resistance to chemotherapy and poor prognosis in various cancer types [33]. We analyzed the mRNA expression levels of *Hsp70 (HSPA1A)* in normal and tumoral breast tissues using two published databases, the Gene Expression database of Normal and Tumor tissues (GENT2) (Figure 1A, left panel) and Gene Expression Profiling Interactive Analysis (GEPIA) (Figure 1A, right panel). As shown, breast tumor samples exhibited significantly higher *Hsp70* mRNA expression levels compared to corresponding normal tissues (Figure 1A). We then examined and analyzed the correlation between *Hsp70* expression and clinical progression in breast cancer patients using the Kaplan–Meier plotter database [34]. Kaplan–Meier analysis showed that Hsp70 expression was negatively associated with overall survival (*p* = 9.2 × 10^−4^). Patients with higher *Hsp70* mRNA expression had shorter overall survival time than those with lower *Hsp70* expression (Figure 1B). In addition, the protein expression level of Hsp70 was higher in human breast cancer cell lines, including TNBC cell lines, than in MCF-10A normal mammary epithelial cells (Figure 1C) [31]. Taken together, these results suggested that Hsp70 expression is overexpressed in breast cancer and negatively associated with clinical breast cancer prognosis. Therefore, it shows that Hsp70 acts as a clinically relevant target for breast cancer treatment.

### 3.2. Cytotoxic and Antiproliferative Activities of Compound 1 and Compound 6 in TNBC Cells

Pursuant to the previous study, we have developed two novel rhodacyanine-based Hsp70 inhibitors, represented by compound 1 and compound 6 (Figure 2A), that effectively inhibited the chaperone activity of Hsp70 and suppressed the expression of Hsp70’s client proteins [31]. We first investigated the cytotoxic activities of compounds 1 and 6 in four TNBC cell lines, and cell viability was determined by MTT assays. As shown, compound 1 (Figure 2B) and compound 6 (Figure 2C) significantly suppressed cell viability in a dose dependent manner in MDA-MB-231, HCC-1937, MDA-MB-468, and HCC-1806 cells. 

To shed light on the efficacy of Hsp70 inhibitors against TNBC cells growth, we further assessed the effect of compounds 1 and 6 on cell proliferation by colony formation assays. Compared to the control treatment of DMSO, both compound 1 and compound 6 significantly reduced colony formations in a dose dependent manner in MDA-MB-231, HCC-1937, and MDA-MB-468 cells (Figure 3A,B). Among the three TNBC cell lines, MDA-MB-468 cells showed the most sensitivity to Hsp70 inhibitors with 73% and 55% inhibition of colony formation at 0.025 μM by compound 1 and compound 6, respectively. These results suggested that inhibition of Hsp70 leads to suppression of TNBC cell growth. The role of Hsp70 in TNBC cell growth was further confirmed by genetic knockdown of Hsp70. We used three different shRNAs (#589, #650, and #755) to reduce Hsp70 expression, and two stable Hsp70 knockdown clones (shHsp70 #650 and shHsp70 #755) were selected to examine the consequent effects on colony formation in MDA-MB-231 cells (Figure 3C). Consistent with the effect of compounds 1 and 6 treatment, knockdown of Hsp70 could significantly suppress the colony forming ability of MDA-MB-231 cells (Figure 3D), suggesting that Hsp70 is involved in the proliferation of TNBC cells and targeting Hsp70 could be a promising option in the treatment of TNBC.

### 3.3. Suppressive Effect of Hsp70 Inhibitors on BCSCs in TNBC

As selective targeting BCSCs is a promising therapeutic strategy to abolish the progression, reduce the risk of recurrence, and improve patient outcome of breast cancer [35], we assessed the effects of compound 1 and compound 6 on mammosphere formation, a surrogate measure of CSC expansion [36], in two TNBC cell lines, MDA-MB-231 and MDA-MB-468. The mammosphere formation assay showed that compound 1 (Figure 4A) and compound 6 (Figure 4B) exhibited potent dose-dependent suppressive effects on mammosphere formation in both cell lines. Moreover, knockdown of Hsp70 in MDA-MB-231 cells, using two different shRNAs (#650 and #755), decreased the ability of MDA-MB-231 to form mammospheres as compared to control cells (Figure 4C), which was reminiscent of the effects observed with compound 1 and compound 6 treatment. Since CSCs have been found to contribute to tumor metastasis, we also examined the effect of Hsp70 inhibition by genetic knockdown on cancer cell motility. The transwell migration assays revealed that the migration abilities were also greatly decreased by Hsp70 knockdown (shHsp70 #650 and shHsp70 #755) compared with control shRNA cells (Figure 4D). The effect of Hsp70 inhibition by compound 1 and compound 6 on BCSCs in TNBC cells was also verified by reduced expressions of the BCSCs-related markers CD133, CD44, and Kruppel-like factor 4 (KLF4) in MDA-MB-231 and MDA-MB-468 cells (Figure 4E). In addition, knockdown of Hsp70 in MDA-MB-231 cells led to concomitant decreases in the expression of CD133, CD44, and KLF4 (Figure 4F). These results indicated that Hsp70 plays a vital role in the regulation of the BCSC populations in TNBC cells.

### 3.4. Hsp70 Inhibition Downregulates β-catenin and its Downstream Target Genes

The Notch and Wnt/β-catenin signaling pathways have been implicated in the regulation of self-renewal and survival of BCSCs [37]. To interrogate the mechanistic link between Hsp70 and BCSCs, we assessed the effects of the Hsp70 inhibitors compound 1 and compound 6 on the expressions of Notch1 and β-catenin in TNBC cells. Western blot analysis indicated that Hsp70 inhibitor-induced suppression of BCSCs in TNBC cells was associated with the inhibition of Wnt/β-catenin signaling, as manifested by parallel decreases in the expression levels of β-catenin and its downstream target genes, NFκB and Cyclin D1, in MDA-MB-231, MDA-MB-468, HCC-1937, and HCC-1806 cells (Figure 5A,B). In contrast, there were no appreciable changes in Notch1 expression in response to compound 1 and compound 6 in MDA-MB-231 and MDA-MB-468 cells (Figure 5C), indicating that Notch1 was not involved in Hsp70 inhibitor-mediated suppression of BCSCs in TNBC cells. The ability of Hsp70 to regulate β-catenin expression was confirmed by shRNA-mediated knockdown of Hsp70 in TNBC cells. Consistent with our premise, knockdown of Hsp70 was associated with reduced expression of β-catenin in MDA-MB-231 cells (Figure 5D).

### 3.5. Evidence That Hsp70 Regulates BCSCs in TNBC via Modulating the Expression of β-catenin

The causal relationship between Hsp70 and β-catenin in maintaining BCSCs in TNBC cells was further confirmed by ectopic expression of β-catenin in Hsp70 inhibitors-treated TNBC cells. As shown, MDA-MB-231 cells were transiently transfected with a β-catenin construct and control vector and the expression of β-catenin was demonstrated by Western blot analyses (Figure 6A,B). In addition, the mammosphere formation assay showed that overexpression of β-catenin abrogated the inhibitory effect of compound 1 (Figure 6A) and compound 6 (Figure 6B) on mammosphere formation in MDA-MB-231 cells, indicating that Hsp70 regulates BCSCs in TNBC cells, at least partially, via modulating the expression of β-catenin.

### 3.6. Hsp70 Inhibition via Pharmacological Inhibitors of Hsp70 or Genetic Knockdown Hampers MDA-MB-231 Tumorigenicity and Tumor Growth

In order to further confirm the role of Hsp70 implicated in regulating BCSCs, compound 1, compound 6, and the isolated stable Hsp70 knockdown clone (shHsp70 #755) of MDA-MB-231 cells were used to investigate the in vivo efficacy of Hsp70 inhibition in suppressing tumorigenesis and xenograft tumor growth, respectively. Since CSCs possess strong tumor-initiating capacity, we examined the effect of compound 1 and compound 6 on tumor initiation by injecting 1 × 10^3^ living cells, which were collected from 2.5 μM compound 1 or compound 6 treated MDA-MB-231 cells for 48 h, versus the same numbers of control DMSO treated cells subcutaneously into both flanks of the nude mice (*n* = 6 for each group). After 12 weeks post-injection, the tumor incidence in mice receiving DMSO control group was 83.3% compared to the respective Hsp70 inhibitors treated groups, in which none of mice developed tumor (Figure 7A). In addition, in a separate experiment, the suppressive effect of Hsp70 knockdown on tumor growth was evaluated by injecting nude mice with 1 × 10^6^ shHsp70 #755 or control cells (*n* = 4) (Figure 7B). Consistent with the results on tumor initiation, knockdown of Hsp70 significantly reduced MDA-MB-231 xenograft tumor growth, as indicated by tumor volume, relative to the control (84% inhibition) (Figure 7C).

## 4. Discussion

TNBC, representing about 15 – 20% of all breast cancers, is one of the most challenging cancers to treat. In spite of the available treatment options for TNBC remains limited, significant progress has been made over the past decades toward the development of novel emerging targeted or combination therapies [2,38,39]. Recently, CSCs have been considered the key driver of tumor progression and metastasis, and to be responsible for cancer relapse and drug resistance [40]. BCSCs have been known to contribute to tumor development, recurrence and metastasis of breast cancer, and therapeutic resistance in the clinic [41]. TNBC cells have a greater mammosphere-forming capacity and more aggressive phenotypes, which are associated with the presence of BCSCs, than non-TNBC cells [42,43]. Clinically, histological analyses also revealed that breast cancer tissues from TNBC patients exhibited enriched CD44^+^/CD24^−^ and ALDH1^+^ expression levels compared to non-TNBC tissues [42,43,44]. Accordingly, many therapeutic strategies targeting the CSCs phenotypes have shown inhibitory effects on tumor initiation, therapy resistance, and metastasis in TNBC cells [3].

Three major signaling pathways, including Wnt/β-catenin, Notch, and Hedgehog, have critical roles in regulating the self-renewal and differentiation of BCSCs. A number of studies have suggested that dysregulated Wnt/β-catenin signaling was correlated with breast cancer development and progression [45,46]. Increased Wnt/β-catenin signaling has also been identified as a driving force of TNBC tumorigenesis [47]. Moreover, aberrant activation of Wnt/β-catenin signaling or transcriptional activity of β-catenin has been implicated in the self-renewal activity and invasive potential of BCSCs [48]. Yang Y. et al. demonstrated that suppression of the Wnt/β-catenin signaling pathway notably inhibited BCSCs self-renewal by a natural product [49]. Compared to other cells, the higher level of Wnt/β-catenin signaling pathway contributes to the high invasiveness and drug resistance of BCSCs. Therefore, inhibition of Wnt/β-catenin signaling or transcriptional activity of β-catenin may be a potential option for TNBC therapy by reducing BCSC stemness [3].

Previous reports have shown that Hsp70 is highly elevated in several types of tumors [50], and constitutively high expression of Hsp70 correlates with increased tumor grades and poor prognosis [19]. Additionally, Hsp70 overexpression has also been found to correlate with more aggressive tumor phenotypes, including cancer cells metastasis and therapy resistance [51,52]. Accumulating evidence also revealed that upregulated Hsp70 expression was found in CSC-like cells [53,54,55], and inhibition of Hsp70 reduced the stemness phenotypes in different tumor types [30,55,56]. Therefore, targeting Hsp70 could be an effective strategy for cancer therapy by eliminating both cancer cells and CSCs. In an attempt for developing the novel therapeutic strategy for TNBC, we provide evidence in this study that inhibition of Hsp70 with Hsp70 inhibitors could not only inhibit cancer cell viability and proliferation, but also suppress mammosphere formation through reducing the expression of β-catenin and its downstream target genes, NFκB and Cyclin D1, in TNBC. Moreover, we used the Hsp70 inhibitors treated TNBC cells and a stable Hsp70 knockdown clone of MDA-MB-231 cells to demonstrate the in vivo efficacy of Hsp70 inhibition in suppressing tumorigenesis and xenograft tumor growth. Together, these findings suggest that the inhibition of Hsp70 may represent a potential therapeutic strategy against breast cancer cells and BCSCs in TNBC. Further studies to evaluate the potential clinical role of Hsp70 in TNBC and the development of derivatives of Hsp70 inhibitors with improved efficacy and safety profiles are warranted.

## 5. Conclusions

Substantial evidence has shown that Hsp70 is overexpressed in multiple types of cancers, including breast, colon, liver, prostate, esophagus, and cervix, suggesting its involvement in tumorigenesis. This study further demonstrated that Hsp70 represents a potential anti-CSCs target in TNBC cells. Accordingly, two putative Hsp70 inhibitors, compound 1 and compound 6, exhibited in vitro and in vivo efficacy in suppressing CSCs through the inhibition of β-catenin in TNBC cells. From a therapeutic perspective, compound 1 and compound 6 might help foster new therapeutic strategies for TNBC treatment to eliminate BCSCs by targeting Hsp70. 

## Figures and Tables

**Figure 1 cancers-14-04898-f001:**
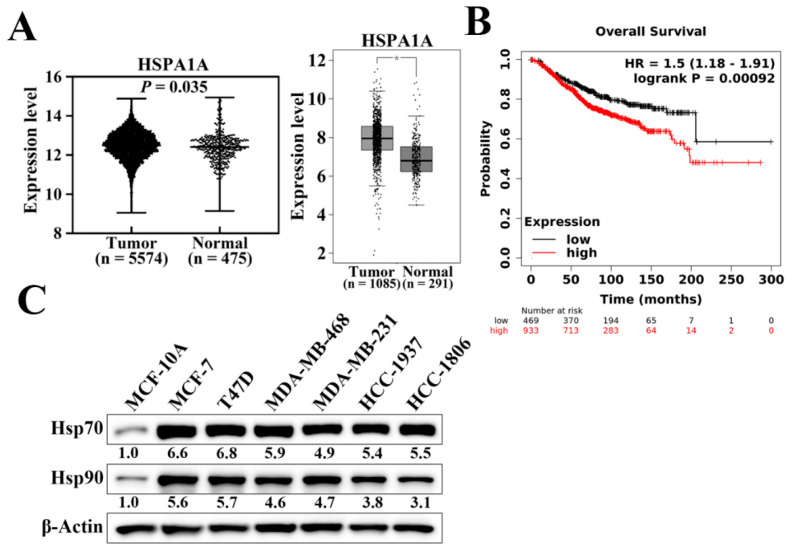
Hsp70 gene and protein expression in clinical breast tissues and breast cancer cells. (**A**) Expression level of Hsp70 in breast cancer (left panel, GENT2 database, *n* = 5574 in cancerous tissue and *n* = 481 in normal tissue, *p* = 0.035; right panel, GEPIA expression database, *n* = 1085 in cancerous tissue and *n* = 291 in normal tissue, * *p* < 0.01). (**B**) Kaplan–Meier analysis of the prognostic value of Hsp70 mRNA expression in the dataset [Affymetrix ID, 202581_at (HSPA1A)]. Survival curves were plotted for all breast cancer patients (*n* = 1402). Cut-off value, median expression level. HR, hazard ratio (with 95% confidence interval). (**C**) Differential expression of Hsp70 and Hsp90 proteins in human breast cancer cell lines versus MCF10A normal human mammary cells. Values under Hsp70 and Hsp90 blots represent the relative expression compared to MCF-10A. The band intensity was measured densitometrically using Image J software and normalized to the β-Actin band.

**Figure 2 cancers-14-04898-f002:**
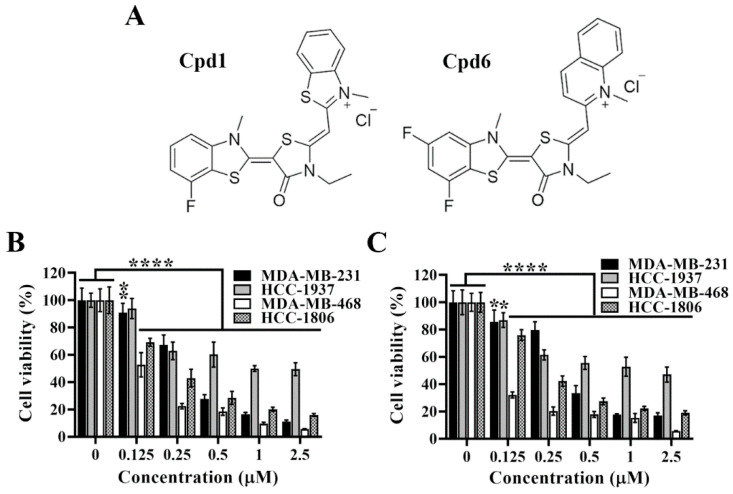
Evidence that compound 1 and compound 6 exhibit high potency in suppressing triple negative breast cancer cells viability. (**A**) Chemical structures of compound 1 and compound 6. (**B**) Dose-dependent suppressive effects of compound 1 and (**C**) compound 6 on the viability of MDA-MB-231, HCC-1937, MDA-MB-468 and HCC-1806 cells after 72 h treatment. Cell viability was determined by MTT assays. Columns, means; bars, S.D., * *p* < 0.05, ** *p* < 0.01, **** *p* < 0.0001.

**Figure 3 cancers-14-04898-f003:**
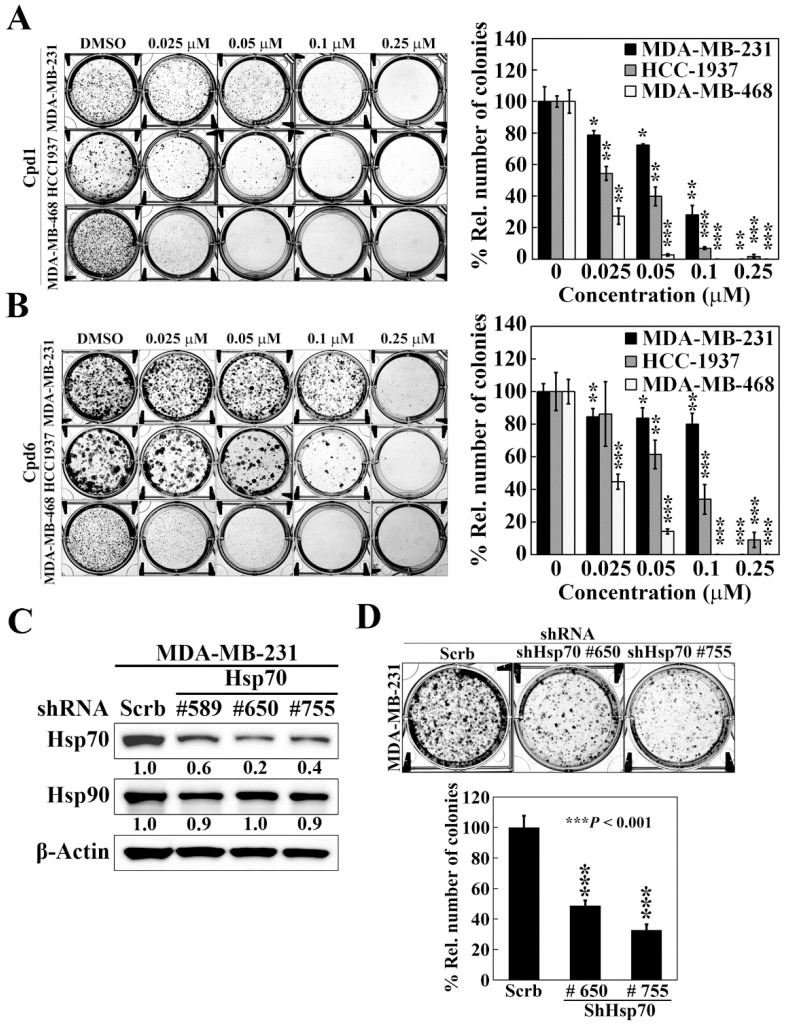
Effects of compound 1, compound 6, and Hsp70 knockdown on cell proliferation in triple negative breast cancer cells. (**A**) Left, representative images showing the effect of compound 1 on colony formation in MDA-MB-231, HCC-1937, and MDA-MB-468 cells. Right, quantitative results of colony formation expressed as the percentage compared with the DMSO control. Data, means ± S.D., * *p* < 0.05, ** *p* < 0.01, *** *p* < 0.001. (**B**) Left, representative images showing the effect of compound 6 on colony formation in MDA-MB-231, HCC-1937, and MDA-MB-468 cells. Right, quantitative results of colony formation expressed as the percentage compared with the DMSO control. Data, means ± S.D., * *p* < 0.05, ** *p* < 0.01, *** *p* < 0.001. (**C**) Western blot analyses of the effect of stable knockdown of Hsp70 on the expression of Hsp70 and Hsp90 in MDA-MB-231 cells. Values under Hsp70 and Hsp90 blots represent the relative expression compared to control shRNA. The band intensity was measured densitometrically using Image J software and normalized to the β-Actin band. (**D**) Upper, representative images showing the effect of stable knockdown of Hsp70 on colony formation in MDA-MB-231 cells. Lower, quantitative results of colony formation from stable knockdown of Hsp70 in MDA-MB-231 cells. Data, means ± S.D., *** *p* < 0.001.

**Figure 4 cancers-14-04898-f004:**
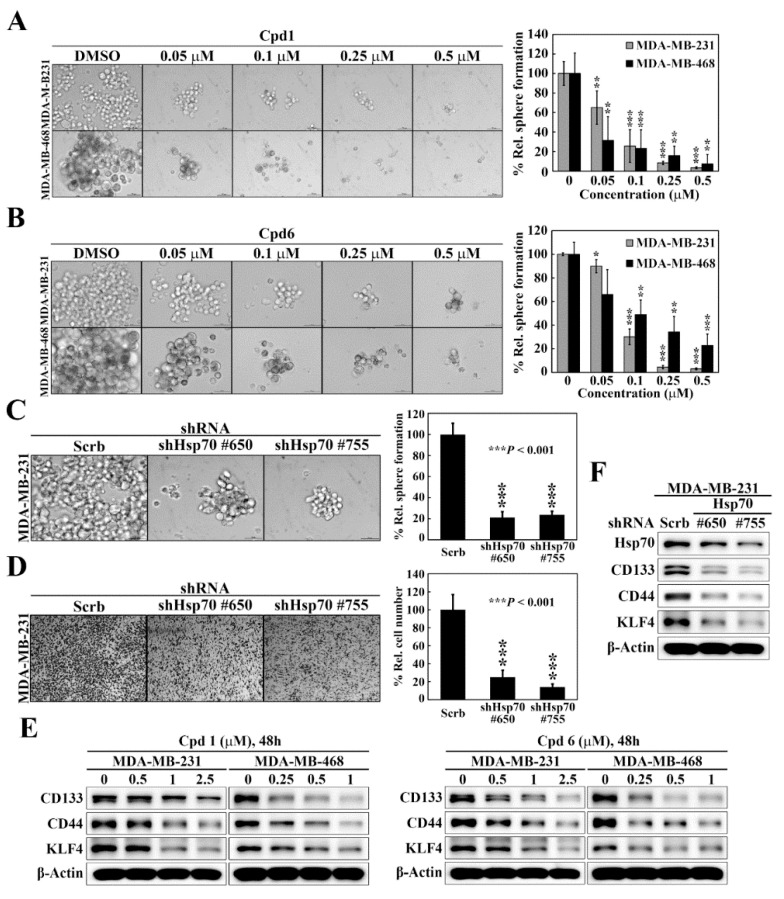
Hsp70 is involved in maintaining cancer stem cells in triple negative breast cancer cells. (**A**) Left, representative images showing the effect of compound 1 on mammosphere formation in MDA-MB-231 and MDA-MB-468 cells (100× magnification). Right, quantitative results of mammosphere formation expressed as the percentage compared with the DMSO control. Data, means ± S.D., ** *p* < 0.01, *** *p* < 0.001. (**B**) Left, representative images showing the effect of compound 6 on mammosphere formation in MDA-MB-231 and MDA-MB-468 cells (100× magnification). Right, quantitative results of mammosphere formation expressed as the percentage compared with the DMSO control. Data, means ± S.D., * *p* < 0.05, ** *p* < 0.01, *** *p* < 0.001. (**C**) Left, effects of stable knockdown of Hsp70 on mammosphere formation in MDA-MB-231 cells (100× magnification). Right, quantitative results of mammosphere formation expressed as the percentage compared with the control shRNA. Data, means ± S.D., *** *p* < 0.001. (**D**) Left, representative images showing the effect of stable knockdown of Hsp70 on cell motility in MDA-MB-231 cells, determined by transwell migration assay. Right, quantitative analyses of the number of migrated cells on the lower side of chambers expressed as the percentage compared with the control shRNA. Data, means; bars, S.D., *** *p* < 0.001. (**E**) Western blot analyses of the effect of compound 1 (left) and compound 6 (right) on the expressions of various cancer stem cell markers, including CD133, CD44, and KLF4 in MDA-MB-231 and MDA-MB-468 cells. (**F**) Western blot analyses of the effect of stable knockdown of Hsp70 on the expressions of cancer stem cell markers, including CD133, CD44, and KLF4 in MDA-MB-231 cells.

**Figure 5 cancers-14-04898-f005:**
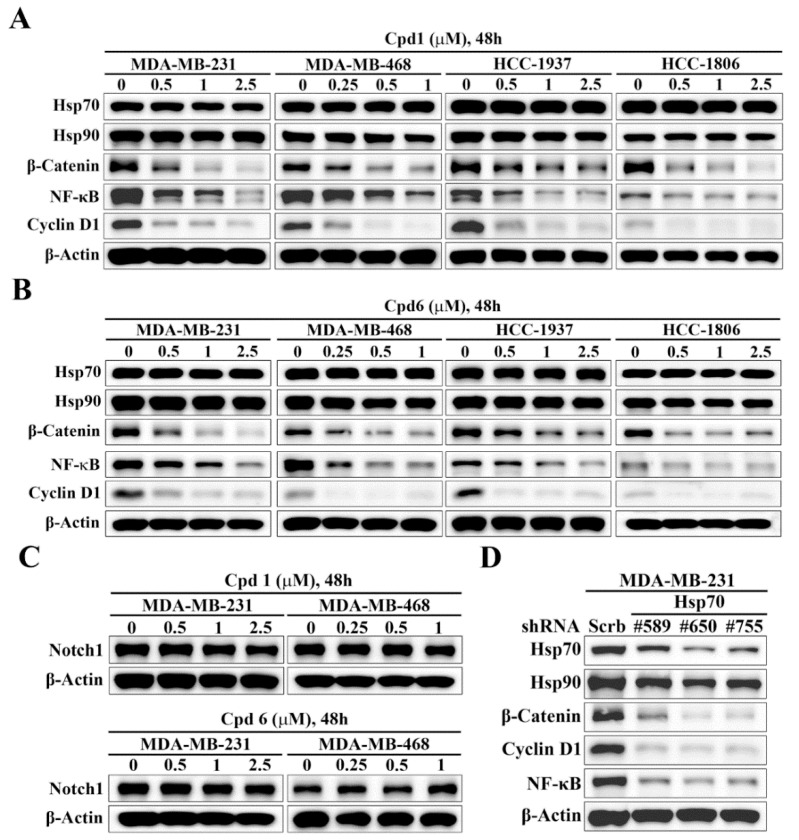
Evidence that Hsp70 regulates the expression of β-catenin and its downstream target genes. Western blot analyses of the effect of (**A**) compound 1 and (**B**) compound 6 on the expressions of Hsp70, Hsp90, β-catenin, Cyclin D1, and NF-κB in MDA-MB-231, MDA-MB-468, HCC-1937, and HCC-1806 cells. (**C**) Western blot analyses of the effect of compound 1 (upper) and compound 6 (lower) on the expression of Notch1 in MDA-MB-231 and MDA-MB-468 cells. (**D**) Western blot analysis of the effect of stable knockdown of Hsp70 on the expressions of Hsp70, Hsp90, β-catenin, Cyclin D1, and NF-κB in MDA-MB-231 cells.

**Figure 6 cancers-14-04898-f006:**
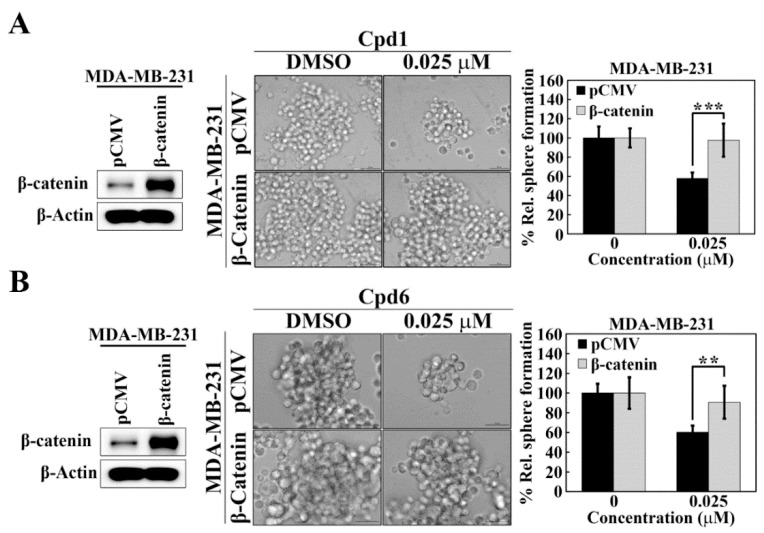
β-Catenin is involved in Hsp70-mediated regulation of cancer stem cells in MDA-MB-231 cells. (**A**) Effects of ectopic expression of β-catenin versus pCMV vector control on mammosphere formation in compound 1 treated MDA-MB-231 cells. Left, Western blot analysis of β-catenin expression in ectopic expression of pCMV vector and β-catenin in MDA-MB-231 cells. Middle, representative images showing the effect of ectopic expression of β-catenin versus pCMV vector control on mammosphere formation in compound 1 treated MDA-MB-231 cells (100× magnification). Right, quantitative results of mammosphere formation. Data, means ± S.D., *** *p* < 0.001. (**B**) Effects of ectopic expression of β-catenin versus pCMV vector control on mammosphere formation in compound **6** treated MDA-MB-231 cells. Left, Western blot analysis of β-catenin expression in ectopic expression of pCMV vector and β-catenin in MDA-MB-231 cells. Middle, representative images showing the effect of ectopic expression of β-catenin versus pCMV vector control on mammosphere formation in compound 6 treated MDA-MB-231 cells (100× magnification). Right, quantitative results of mammosphere formation. Data, means ± S.D., ** *p* < 0.01.

**Figure 7 cancers-14-04898-f007:**
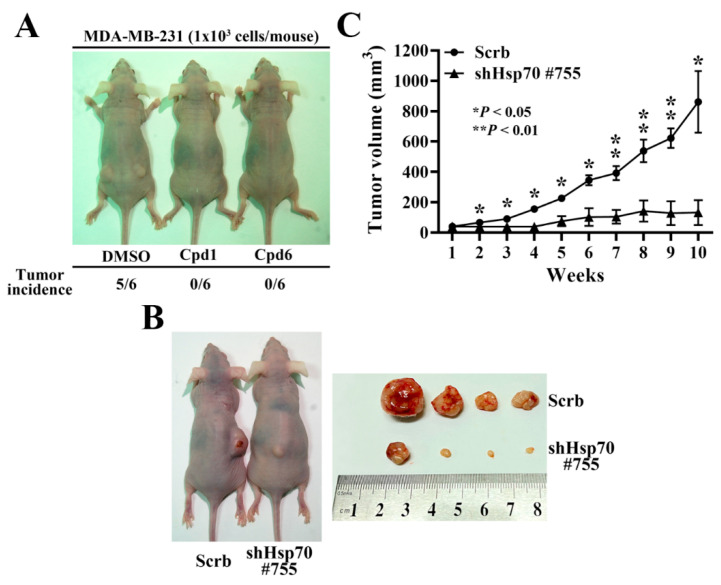
In vivo efficacy of Hsp70 inhibitors and stable Hsp70 knockdown in suppressing MDA-MB-231 tumorigenicity and xenograft tumor growth in nude mice. (**A**) Comparison of the tumor incidence between Hsp70 inhibitors treated and control DMSO treated MDA-MB-231 cells. 1 × 10^3^ DMSO, compound 1, or compound 6 treated MDA-MB-231 cells were subcutaneously injected into both flanks of the mice. (**B**) Representative images of stable control knockdown (Scrb) and Hsp70 knockdown clone (shHsp70 #755) on subcutaneous MDA-MB-231 xenograft tumor-bearing mice (left) and dissected tumor samples (right). (**C**) Suppressive effect of Hsp70 knockdown (shHsp70 #755) on subcutaneous MDA-MB-231 xenograft tumor volumes (means ± S.E.M., *n* = 4). * *p* < 0.05, ** *p* < 0.01.

## Data Availability

The data presented in this study are available upon reasonable request.

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
