# Peer review of "Targeting Triple Negative Breast Cancer Stem Cells by Heat Shock Protein 70 Inhibitors"

_cancers, 2022, doi:10.3390/cancers14194898_

Round 1

Reviewer 1 Report

The manuscript describes that two Hsp70 inhibitors, compound 1 and compound 6, are specifically cytotoxic against different breast cancer stem cells. These results are sound and interesting and this reviewer thinks that they merit publication in Cancers provided the following concerns have been amended/clarified.

MAJOR CONCERN

1.      Authors have investigated in nude mice the effect of their inhibitors on tumorigenicity. In vivo experiments with compounds 1 and 6 are not very convincing.

A-     To carry on these experiments, they have treated the cells with 2.5 uM of the compounds for 48h. The treatment of these cell lines with this concentration for 72h produce the death of 90% of the cells (Figures 2 and 3).  Authors must clarify how this concentration affects cell growth in vitro at 48h. Why they did not use lower concentrations of the compounds?

B-     In these experiments, authors have injected 1000 cells, but when investigating the effect in vivo of the genetic knockdown of hsp70 they inject 1,000,000 cells. As a result, the size of the tumors are clearly higher in the latter case (compare Figures 7A and 7B). Why this difference in the protocol? Can the authors discard that they would not had observed the appearance of tumors if a higher number of cells (106 cells) were inoculated?

C-     It is not clear in the manuscript how, after the treatment with compounds 1 and 6, these cells were collected. Did the authors remove the dead cells from the samples? This must be clarified in the draft.

MINOR CONCERNS

2.      Most of the results presented in Figure 1-C have been also reported for the authors in their Current Medicinal Chemistry article (Figure 2-B). Authors must indicate that they results are in accordance with previously described (32).  

3.      The sentence between  lines 237 and 240 should be rewritten. May be by introducing a point between Hsp70 and “using” in line 238?  

Reviewer 2 Report

Dear authors, the manuscript is well written and very interesting in the field of cancer research.

Breast cancer is one of the most frequent causes of death in women worldwide.

Although research in recent years has developed better diagnostic methods and developed targeted therapy, the emergence of resistance after chemotherapy treatment is a major problem in the care of breast cancer patients.

Therefore, the study of new therapeutic targets is very important for the overall survival of patients. Heat shock protein, which is involved in many cellular processes, could be one of them, particularly HSP70, whose overexpression has been demonstrated in many tumour types, including triple-negative subtype breast cancer.

This work will be suitable for publication after a major review:

1. Why are only 72 hours of treatment with CPD 1 and 6 being studied? What happens at 24-48 hours?

2. Regarding the statistical analysis of survival, the A-Nova test could be performed instead of the t-Student test.

3. Figure 1 panel C, add a histogram with the relative expression of HSP70 and HSP90.

4. Was cytotoxicity of CPDs 1 and 6 performed in MCF-10A? If the answer is none, can it be done?

5.  Figure 3 panel C, add a histogram of the expression of HSP70 and HSP90 after silencing.

6. In line 66, add that "HSPs participate in repair processes" by including a new reference " Thin ML, Nadin SB. Heat shock proteins and DNA repair mechanisms: an updated overview. Cell Stress Chaperones. 2018 May;23(3):303-315. doi: 10.1007/s12192-017-0843-4. Epub 2017 Sep 26. PMID: 28952019; PMCID: PMC5904076."

7. In line 69 add after overexpression of HSP70 add "analogous to two proteins involved in the nucleotide excision repair (NER) pathway, Cockayne symdrome group A and B protein" (refs. Caputo M, Frontini M, Velez-Cruz R, Nicolai S, Prantera G, Proietti-De-Santis L. CSB repair factor is overexpressed in tumor cells, increases apoptotic resistance and promotes tumor growth. DNA Repair (Amst). 2013 Apr 1;12(4):293-9. doi: 10.1016/j.dnarep.2013.01.008. Published February 16, 2013. PMID: 23419237; PMCID: PMC3610032; Filippi S, Paccosi E, Balzerano A, Ferretti M, Poli G, Taborri J, Brancorsini S, Proietti-De-Santis L. CSA Antisense Targeting Enhances Anticancer Drug Sensitivity in Breast Cancer Cells, Including the Triple-Negative Subtype. Cancers (Basel). 2022 Mar 26;14(7):1687. doi: 10.3390/cancers14071687. PMID: 35406459; PMCID: PMC8997023) and then to continue with.

has been observed in several forms of cancer, including breast, colon, liver, prostate, esophagus, and cervix.

Reviewer 3 Report

This paper shows an interesting activity of two compounds (named 1 and 6) on the proliferative activity of TNBC cells, as a result of the inhibition of HSP70. This correlates with the decrease of expression of Beta-catenin. Theresults are well presented and are convincing, however I have two major concerns about the experimental plan.

a) in any of the experiments in which there a treatment with compounds 1 and 6 there is a control experiment including MCF-10a cells

b) the effects on animals in which cancer cells are implanted after the treatment with these compounds should be complemented with a new one in which the treatment follows the tumor implant

Round 2

Reviewer 1 Report

I think that the ms. can be published in Cancers in its present version

Reviewer 2 Report

Dear Authors, now the manuscript is elligible for the pubblication